# Antimicrobial Resistance Profile by Metagenomic and Metatranscriptomic Approach in Clinical Practice: Opportunity and Challenge

**DOI:** 10.3390/antibiotics11050654

**Published:** 2022-05-13

**Authors:** Langgeng Agung Waskito, Yudith Annisa Ayu Rezkitha, Ratha-korn Vilaichone, I Dewa Nyoman Wibawa, Syifa Mustika, Titong Sugihartono, Muhammad Miftahussurur

**Affiliations:** 1Department of Internal Medicine, Faculty of Medicine, Universitas Airlangga, Surabaya 60132, Indonesia; langgengaw@gmail.com; 2Helicobacter pylori and Microbiota Study Group, Institute of Tropical Diseases, Universitas Airlangga, Surabaya 60115, Indonesia; yudithannisaayu@gmail.com; 3Department of Physiology and Medical Biochemistry, Faculty of Medicine, Universitas Airlangga, Surabaya 60132, Indonesia; 4Department of Internal Medicine, Faculty of Medicine, Universitas Muhammadiyah Surabaya, Surabaya 60115, Indonesia; 5Gastroenterology Unit, Department of Medicine, Faculty of Medicine, Thammasat University Hospital, Khlong Nueng 12120, Pathumthani, Thailand; vilaichone@hotmail.co.th; 6Digestive Diseases Research Center (DRC), Thammasat University, Khlong Nueng 12121, Pathumthani, Thailand; 7Department of Medicine, Chulabhorn International College of Medicine (CICM), Thammasat University, Khlong Nueng 12121, Pathumthani, Thailand; 8Division of Gastroentero-Hepatology, Department of Internal Medicine, Faculty of Medicine, Dr. Soetomo Teaching Hospital, Universitas Airlangga, Surabaya 60286, Indonesia; titongsppd@gmail.com; 9Division of Gastroentero-Hepatology, Department of Internal Medicine, Sanglah General Hospital, Faculty of Medicine, Universitas Udayana, Denpasar 80232, Indonesia; agusbobwibawa@yahoo.com; 10Division of Gastroentero-Hepatology, Department of Internal Medicine, Dr. Saiful Anwar Hospital, Malang 65112, Indonesia; drtika_78@ub.ac.id

**Keywords:** metagenomic, metatranscriptomic, antibiotics resistance, bioinformatics, clinical practice

## Abstract

The burden of bacterial resistance to antibiotics affects several key sectors in the world, including healthcare, the government, and the economic sector. Resistant bacterial infection is associated with prolonged hospital stays, direct costs, and costs due to loss of productivity, which will cause policy makers to adjust their policies. Current widely performed procedures for the identification of antibiotic-resistant bacteria rely on culture-based methodology. However, some resistance determinants, such as free-floating DNA of resistance genes, are outside the bacterial genome, which could be potentially transferred under antibiotic exposure. Metagenomic and metatranscriptomic approaches to profiling antibiotic resistance offer several advantages to overcome the limitations of the culture-based approach. These methodologies enhance the probability of detecting resistance determinant genes inside and outside the bacterial genome and novel resistance genes yet pose inherent challenges in availability, validity, expert usability, and cost. Despite these challenges, such molecular-based and bioinformatics technologies offer an exquisite advantage in improving clinicians’ diagnoses and the management of resistant infectious diseases in humans. This review provides a comprehensive overview of next-generation sequencing technologies, metagenomics, and metatranscriptomics in assessing antimicrobial resistance profiles.

## 1. Introduction

Resistance to antibiotics is a major burden worldwide. The burden is affecting not only the health service but also the economic sector. The increasing risk of severe infection combined with the growth of antimicrobial resistance (AMR) has become a huge concern of the World Health Organization (WHO), European Centre for Disease Prevention and Control, and the United States (US) Centers for Disease Control and Prevention (CDC). Currently, >700,000 individuals die due to antibiotic resistance, including 230,000 patients with tuberculosis infection [1,2]. Further, the WHO predicted that antibiotic resistance could cause 10 million deaths annually by 2050 [3]. From the economic perspective, it was estimated that antibiotic resistance would cost as much as USD 20 billion in direct costs combined with USD 35 billion from productivity lost due to illness in the USA [4]. Moreover, in the European Union and European Economic Area countries, an increase in healthcare cost and productivity loss of EUR 1.1–1.5 billion annually is expected due to antibiotic resistance infection if appropriate and effective actions are not executed [5].

A recent systematic review showed that dysbiosis related to inflammatory bowel diseases increased the relative abundance of several bacteria and estimated that only <30% of the gut microbiome was culturable [6]. Furthermore, the current widely used approach to examine antimicrobial susceptibility relies on a culture-based examination. Some gastrointestinal (GI) pathogens might be cultured with ease, such as *Escherichia coli*, *Klebsiella* sp., and *Staphylococcus* sp. However, gastric pathogens, including Helicobacter pylori, are slightly difficult and need more time to culture. These conditions lead to difficulties in managing GI infections. Therefore, it is necessary to implement other approaches in the determination of antibiotic resistance, even for uncultured bacteria.

In the last decade, the advancement of next-generation sequencing (NGS) technology has shown remarkable results, enhancing shifts in paradigms in disease management. The application of metagenomic studies in non-infectious diseases, such as malignancy and degenerative diseases (e.g., diabetes mellitus, hypertension, chronic kidney disease, and Alzheimer’s disease) could elucidate the associated bacteria [7,8,9,10]. Specifically, in GI diseases, this particular approach also showed the role of the microbiome in the development of GI tract diseases as well as a potential marker for antibiotic resistance [11,12]. Additionally, metatranscriptomic studies focusing on RNA sequencing (RNA-seq) also make it possible to discover functional shifts in certain GI diseases [12,13]. Moreover, the bioinformatics-based technique has advantages for discovering new antibiotics by studying the existence of antibiotic resistance genes (ARGs) and the resistome [14,15]. Therefore, the utilization of NGS technology to accelerate the determination of antibiotic resistance for the appropriate choice of antibiotic therapy and drug discovery is an interesting research subject.

This review discusses the application of NGS technology using metagenomic and metatranscriptomic approaches to detect antibiotic resistance. Since this topic covers a wide research field, this review limits it to the GI tract. This review also elaborates on the advantages and current challenges in the implementation of this approach in the clinical routine.

## 2. Global Burden of Human Gut-Resistant Pathogens

Antimicrobial resistance is a predictable matter to the point of certainty. The advent of super resistance in microbes in the last 25 years has altered the condition into a life-threatening issue. The US CDC estimated that of 2 million infected patients, 23,000 will die due to AMR bacterial infections [3]. In addition, the CDC has categorized 18 multidrug-resistant (MDR) pathogens as urgent, serious, and concerning threats. This emphasizes the urgency of the AMR threat in the context of infectious diseases, including GI tract infections. Diarrheal pathogen infections can also be caused by potent GI tract infections [16]. These horrifying bacteria include drug-resistant non-typhoidal *Salmonella*, extended-spectrum beta-lactamase (ESBL) Enterobacteriaceae, carbapenem-resistant Enterobacteriaceae, and drug-resistant *Shigella*, which are also listed by the CDC as urgent MDR pathogens [3]. In addition, non-typhoidal *Salmonella* sp. was reported to be resistant to ampicillin and fluoroquinolone antibiotics for diarrhea. Further, *Campylobacter* sp. has recently developed resistance to major antibiotics, including ciprofloxacin, ampicillin, erythromycin, tetracycline, and trimethoprim–sulfamethoxazole [17]. Moreover, ESBL-producing Enterobacteriaceae have been reported to be resistant to several antibiotics, including third-generation cephalosporins, penicillin and its derivatives, folate pathway inhibitors, and fluoroquinolones [17,18]. These data emphasize the emergence of the AMR threat in GI tract infection cases.

Several mechanisms are hypothesized to explain the development of AMR bacteria, including intrinsic, acquired, and adaptive mechanisms [19,20]. Intrinsic resistance is defined as an innate ability of bacteria to resist specific antibiotics and is not associated with unfavorable conditions related to antibiotic exposure. Acquired resistance is described as a result of chromosomal mutations or gene acquisition through mobile genetic elements (MGEs) by horizontal gene transfer [21]. The acquisition of MGEs occurs via several mechanisms, including conjugation, transduction, and transformation [22]. Lastly, adaptive resistance is still an unclear mechanism and under ongoing investigation. Usually, at a non-bactericidal antibiotic concentration, bacteria can still grow normally, although it is possible to induce mechanisms for using antibiotics and/or related ones [23]. Through a combination of these mechanisms, bacteria resistant to antibiotics usually alter several bacterial functions related to antibiotics, such as reducing bacterial cell permeability, increasing the activity of efflux pump systems, synthesizing some enzymes that could reduce drug activity, performing some modifications, substituting or disrupting antibiotics’ bacterial targets, and forming biofilms [24].

## 3. Current Clinical Routines for AMR Detection

Profiling antimicrobial-resistant organisms is an extremely critical matter in healthcare, especially for infection cases. The accurate susceptible/resistant determination of the infectious pathogen may lead to a shorter length of stay, better clinical outcomes, and lower costs as a direct result of using the right choice of antibiotics [23,24]. Current widely used methods for determining antibiotic-resistant organisms rely on the culture-based approach for culturable bacteria with various setups, including broth and agar dilution assays, agar disk diffusion, and E-test.

Although it has been routinely performed in clinical practice, there are several disadvantages and/or confounding factors of AMR identification using culture-based methods. First is the inability to identify the AMR of bacteria that are in a viable but nonculturable (VBNC) state. Potential pathogens can be in the VNBC state, including *E. coli*, *Salmonella*, and *H. pylori* [25,26,27]. It was discovered that both *E. coli* and *Salmonella* are capable of returning to the actively metabolizing state with the potential to initiate diseases [25,27]. As for *H. pylori*, the VBNC state is known to have a coccoid form, and the ability to return to the pathogenic state with a spiral shape remains insufficiently described. In environmental specimens, such as wastewater or soil, disinfection methods, including chlorination, are reported to reduce the ability to culture *E. coli* and *Salmonella typhimurium* by >99%. As for the clinical scenario, empirical antibiotic therapy may also induce the VBNC state of many pathogenic bacteria [28]. This confounding factor could be overcome with more steps in the subculturing process in the cultivation state. Second, the fact that only a small proportion of environmental microbes are cultured under laboratory conditions is a potential bias for AMR assessment. In addition, the cultured bacteria’s genetic features related to AMR might also misrepresent the microbial community since the gene only represents the cultured bacteria. Due to this bias of the culture-based approach in AMR examination, resistance data should be interpreted. It is well known that the existence of a gene in the environment might not represent the functionality or activity of the gene itself. Hence, the gene activity represented by expression measurement is a necessity, and gene-related AMR exists in the microbial community. Therefore, there are some roles of the molecular-based method in identifying AMR to close the gap of the culture-based approach.

## 4. Overview of Metagenomic and Metatranscriptomic Research

Metagenomics refers to the characterization of all available DNA in a certain environment. This DNA can be isolated from organisms or free-floating DNA. Characterization of human microbiota can be performed in anybody site, including the skin, oral mucosa, lungs, and digestive tract (esophagus, stomach, and gut). The most widely studied body site in microbiota studies is the human gut, which is considered the “human second brain” since its ability to perform complex functional processes in the human body is similar to that of the human brain [29]. Several molecular-based techniques are available to characterize the microbial community in a certain environment. The available techniques with their advantages and disadvantages are summarized in Table 1. The culture approach is a considerably obsolete approach since it also does not have good sensitivity, with only less than 30% of gut microbes being cultured [29]. In recent years, approaches for identifying microbes using NGS, such as sequencing of 16s rDNA and shotgun sequencing, are able to quantitatively measure microbes and quickly identify novel microbes. Since the human microbiota field is extremely wide, we only focus on the human gut and gastric microbiota.

The application of a human microbiota study mostly characterizes microbes in certain conditions, such as infection or cancer. Hence, it could reveal the microbial compositions of certain diseases. The imbalance in the microbial composition that leads to a functional shift is defined as dysbiosis. Dysbiosis usually reduces the microbial diversity with the association of one microbe or group of microbes responsible for dysbiosis [30]. The characterization of human gut microbiota has shown several remarkable findings toward the definition of human physiological functions. It has been described to have several axes toward different systems and functions, including the cardiovascular system, neurology, human reproduction, immunology, and carcinogenesis [31,32,33]. These findings suggest the current trend of microbiota studies to find different microbial communities in various human health conditions and novel pathogens associated with the development of diseases.

A metagenome study goes beyond the sequence of microbes’ DNA in a certain environment. This study also analyzes the sequence of DNA that might not belong to any organism. This wider approach allows us to find MGEs in the environment. Compared to a microbiome study, it has its own technical difficulties since MGEs might only be a fraction of the available DNA in the environment. These MGEs have different functions in the adaptation of bacteria in unfavorable conditions, and these elements can exist in many conditions [34]. Hence, the technical aspect of preparing specimens prior to sequencing is more challenging. Studies on *H. pylori* showed that the MGE’s distribution was different among various population genetics, with an association with particular *H. pylori* phylogeographic lineages, and further provided evidence on both contemporary interlineage and interspecies transfer and displacement [34,35]. One study described that the MGE characteristics of infants are extremely similar to those of their mother on the basis of several factors, and breastfeeding termination and the use of antibiotics were associated with higher abundances of specific ARGs [36]. These findings confirmed the importance of MGEs in bacterial evolution and in sculpting the genome characteristics, including resistance to antibiotics.

Metatranscriptomic research has a different description from metagenome research. While the metagenome only describes the available DNA in the environment, the metatranscriptome explores the full range of messenger ribonucleic acid (RNA) molecules expressed by an organism [14]. Although it is technically more difficult to conduct this study because RNA samples are harder to handle than DNA samples, the nature of the data from the functional profiling of cells provides a better understanding of biological processes [37]. Hence, it can offer important information on the significant biological processes behind the maintenance of the cells.

There are three different approaches for exploring messenger RNA in cells: hybridization-based analysis (microarray technology), Sanger sequencing–based techniques (e.g., serial analysis of gene expression (SAGE) and massively parallel signature sequencing (MPSS)), and the high-throughput NGS approach (RNA-seq) [38]. Differences in the techniques, advantages, and disadvantages of these methodologies are summarized in Table 2. Microarray hybridization–based transcriptome analysis typically involves the incubation of fluorescently stained cDNA together with custom-made microarrays or commercial high-density oligonucleotide microarray probes [39]. Generally, hybridization-based approaches can produce massive data but are relatively cheap, except for large-genome characterization that requires higher-resolution expression [38]. Nevertheless, some limitations exist in this approach, including the dependence on the known genome sequence and potentially high background level due to cross-hybridization reactions [40]. Tag-based methods, including SAGE, cap analysis of gene expression, and MPSS, were introduced to overcome these limitations of the hybridization-based approach. They offer high throughput and more precise “digital” expression levels with a lower background level [41]. However, due to the Sanger sequencing–based examination, it can be more expensive than hybridization-based methods, and some tags might be unable to uniquely map to the reference genome. Further, only part of the transcript is analyzed, and similar sequences are indistinguishable from each other [38].

These disadvantages restrict the utilization of traditional sequencing technology in annotating the structure of transcriptomes. RNA-seq is the most recent approach to characterizing gene expression using deep sequencing technology. This approach offers several advantages over other techniques, including the ability to detect novel transcription factor alterations, which might not be detected by probe-based detection methods. Furthermore, the dynamic range for transcriptomic alterations is extremely wide, up to >8000-fold; hence, it is able to describe a better resolution of transcriptomic changes [42]. Moreover, due to the same platform being used to obtain sequence data, it is easier to obtain and compare our data to other research data, even with some concerns about the normalization step for analysis [43]. Although it offers these advantages, the major disadvantage of this approach is the inability to differentiate between messenger RNA (mRNA), microRNA (miRNA), noncoding RNA, and other RNAs due to the nature of sequencing of the available RNA. In addition, larger RNA molecules are different from small RNAs, such as PIWI-interacting RNAs (piRNAs), short interfering RNAs (siRNAs), microRNAs (miR-NAs), and many other RNAs that can be directly sequenced after the adaptor attachment process. Large RNAs need to be fragmented into smaller sizes, usually approximately 200–500 bp, to be compatible with sequencers, mostly the Illumina platform sequencer. There are several common fragmentation techniques, including RNA fragmentation (RNA hydrolysis or nebulization) and cDNA fragmentation (DNase I treatment or sonication) [44,45].

**Table 2 antibiotics-11-00654-t002:** Comparison of approaches to characterize expression level.

Characteristic	Technology		
	Microarray	cDNA or EST Sequencing	RNA-seq
Detailed specification [46,47,48]			
Basic examination principle	Hybridization	Sanger sequencing	High-throughput sequencing
Genome sequence dependency	Yes	No	Some cases
Resolution	From several to 100 bp	Single base	Single base
Output results	High	Low	High
Background noise	High	Low	Low
*Application* [49,50]			
Characterized gene expression	Yes	Limited for gene expression	Yes
Quantification range	Up to a few hundredfold	Not practical	>8000-fold
Differentiation of isoform ability	Limited	Yes	Yes
Differentiation of allelic expression	Limited	Yes	Yes
*Practical issues* [51,52]			
RNA requirement	High	High	Low
Cost for large genome	High	High	Relatively low

The RNA sequencing technique to describe alterations in the transcriptome has been conducted in various research fields, including gastroduodenal diseases. One study described the difference in transcriptomic expression between patients with Crohn’s disease (CD) and normal individuals and found that 15 genes were upregulated and 4 were down-regulated [49]. Another study that described the transcription difference between gastric cancer (GC) and normal mucosa showed 114 genes exhibiting significant differential expression patterns, and CDH1 was the most significantly upregulated gene, which was 309-fold higher in cancer samples, while DPT was the most downregulated gene, showing a 40-fold change [46]. Another transcriptome profiling study in Chinese patients with GC revealed 36-fold higher expression of CDH1, while DPT and TGFBR2 showed decreased expression in cancer samples [47]. The low expression of DPT in oral cancer has also been validated by qRT-PCR, which substantiates the role of DPT as a common player in various cancers [48]. It is also able to detect novel long noncoding RNAs (lncRNAs) such as HOXC-AS3, which has a role in regulating cell proliferation and migration. Knockdown preferentially affects genes that are linked to proliferation and migration [50]. Moreover, the lncRNA THAP9-AS1 was upregulated after infection of GC cells with *H. pylori* and was higher in GC tissues than in gastritis tissues [51].

Besides being used to describe host transcriptome differences, the RNA-seq approach can also be applied to describe pathogen transcriptomic alterations, such as *H. pylori*. In the challenging yet natural niche of *H. pylori* of high-acidity environmental conditions, genes encoding for antioxidant proteins, flagellar structural proteins, particularly class 2 genes, T4SS/Cag-PAI, FoF1-ATPase, and proteins involved in acid acclimation were highly expressed at acidic pH [51,52]. Moreover, in high-salt conditions, several transcriptomes, including *sabA*, *hopA*, and *hopQ*, are increased, whereas transcript levels of *fecA2* and *fecA3* are decreased, suggesting that the attachment process of *H. pylori* to gastric mucosa is significantly increased in a high-salt environment [53]. These data confirmed the high flexibility of the characterization of the transcriptome using a deep sequencing technique.

## 5. Profiling AMR Using Metagenomic and Metatranscriptomic Approaches

Antibiotic resistance might not only cause problems in infected patients. A resistant organism that developed through several mechanisms also caused a gut microbial imbalance, which later became a systemic syndrome. This microbial imbalance was reported as the major predisposing factor for vancomycin-resistant enterococci [53,54] and other antibiotic-resistant organisms (AROs), including *Klebsiella* and *Escherichia coli* [55]. As the microbial imbalance is considered the major risk factor causing infections with these organisms, the approach for “rebooting” the healthy gut microbiome is the best consideration. In addition, by using the metagenomic approach, we can investigate the gut microbial community state of ARO-infected individuals compared to non-carriers [56]. Furthermore, the reason that ARO-infected patients develop symptoms or remain asymptomatic and the specific communities and functions that provide colonization resistance against those AROs could also be examined.

The metagenomic approach for profiling bacteria that are susceptible and resistant to antibiotics has tremendous potential. The current conventional approach relies on the identified and isolated organism. However, metagenomic data contain several important components, including antibiotic resistance determinants within the microbial community, known as resistomes [55,57]. By considering this element, it could provide a comprehensive picture of the “potential” antibiotic resistance of a specific site’s microbial community. The resistome is useful in predicting the possible resistance pattern of a certain microbial community [58]. Although it has huge potential, it still relies on the known mechanism of resistance and would not be useful in discovering novel or unknown mechanisms. The key advantage of this approach is that the resistome can be determined even though it is not present in the current disease-causing pathogens but might be horizontally transferred to the bacteria due to highly mobile elements of the genetic determinants [21,59]. In addition, the ARG diversity is highly affected by diet composition. There is evidence that a protein increase and carbohydrate reduction in the diet were associated with increased ARG diversity in the canine gut [60]. These data imply that dietary nutritional content, especially protein content, is associated with the gut resistome. Further, it might become a new approach for combating antibiotic resistance through dietary plans [61].

The human gut microbiota can affect host health by metabolizing antibiotics. Although there can be an alteration in the microbial composition after antibiotic exposure, it remains unclear which microorganisms are actively affecting the biological function and which factors elicit their activity and gene expression [62]. To provide a more refined explanation, the metatranscriptomic approach has some other advantages for describing short-term antibiotic exposure, which can considerably affect the gene expression, physiology, and structure of the microbiome [63]. In *Neisseria gonorrhoeae*, transcriptome profiling showed evidence that genetic distance and population structure affect the transcriptional response to azithromycin with four transcripts (rpsO, rplN, omp3, and NGO1079) that were the most significantly regulated between phenotypes with drug exposure [64]. Another study describing multidrug resistance of *Salmonella enterica serovar typhimurium* showed an alteration in the expression of nearly half of the genes, particularly genes associated with adherence function, and those within *Salmonella* pathogenicity islands were significantly upregulated after exposure to subinhibitory concentrations of both chlortetracycline and florfenicol [65].

The database storing currently known AMR-related genes is extremely important for identifying ARGs. There are several databases, such as the Comprehensive Antibiotics Resistance Database [66] and the Antibiotics Resistance Gene Database [67]. In addition, ARG-ANNOTation includes software containing 1689 curated ARGs without a web-based interface [68]. As for mobile genetic determinant databases, ResFinder and resistance determinants database [69] are handy for the appropriate interpretation of a sample of the resistance level based on the gene content. Another gene database is ResistoMap, a handy interactive tool for the visualization of the identified resistome [61]. Interestingly, the expression of a new gene, LSTrAP Crowd, showed the promising potential of characterizing genes that were transcriptionally associated with protein synthesis genes in common bacterial pathogens and thus provides a resource for potential antibiotic development targets or functions that cause antibiotic resistance [70]. Combining, curating, and developing currently known AMR and resistome databases are extremely necessary to increase the identification reliability.

Metagenomic implementation for profiling AMR in the human gut has been conducted in several places worldwide. The summarized results of discovered AMR genes around the world are presented in Table 3. Vancomycin resistance operons, such as VanRG, have been commonly observed worldwide alongside other AMR genes, including genes related to resistance to bacitracin, tetracycline, bacitracin, the macrolide–lincosamide–streptogramin (MLS) group, and cephalosporin [15,60,61]. Generally, these AMR genes were commonly found in Gram-positive, anaerobic, and host commensal bacteria. The most common AMR observed was tetracycline-related resistance (tetQ gene) [71]. Despite the fact that some remote areas with AMR did not have any exposure to commercial antibiotics, another study suggested that the diversity and abundance of AMR genes in the gut are highly affected by antibiotic use [15,71]. Although the frequency of antibiotic use is correlated with ARG, this is not the case for tetracycline resistance determinants. The use of tetracycline in clinical practice is rare, but tetracycline-related ARGs are the most ubiquitous and commonly observed in the human gut [72]. Several hypotheses explain this peculiar phenomenon. First, the original function of tetracycline resistance genes might not be related to AMR (e.g., protein transport, cell communication, and signal tracking) due to the fact that it is identified even in 30,000-year-old permafrost [73]. Second, the consumption of seafood, rice, and vegetables that might be contaminated by heavy metals caused the co-selection of tetracycline resistance genes [74]. Interestingly, the distribution of AMR genes was also related to the geographic origin. A comparison study between Chinese individuals and those from ten other countries showed that the AMR gene type (MLS) and one subtype (erythromycin resistance gene, ermF) were frequently observed in the Chinese population compared to other populations [60]. Another interesting result involves vancomycin resistance genes for Enterococcus/Streptococcus strains, which are highly similar to those in healthy human gut microorganisms, and it is also observed in permafrost samples [73]. These findings suggest that resistance to vancomycin is an intrinsic feature of human gut microbes, although further investigation is necessary.

## 6. Metagenomic and Metatranscriptomic Approaches in Clinical Practice: Yet an Expensive Hope

The current frequently used application of the metagenomic technique in the clinical scenario is pathogen detection. In several cases of infection of unknown origin (IUO), the conventional test that relies on the “hypothesis-exist” scenario failed to uncover the infectious agents. In contrast, the metagenomic approach has the advantage of the “hypothesis-free” assumption, which has more potential in detecting the “unknown” pathogen. Although this “hypothesis-free” scenario seems promising, one drawback of this approach is the background bias of the human genome, thus limiting the overall analytical sensitivity of the approach for pathogen detection [85]. Targeted sequencing could be a solution to this obstacle. If only bacterial sequences are of interest, then targeted sequencing of the 16S rRNA gene would be able to distinguish most species while, not incidentally, sequencing the human host background [82]. As for the clinical setting, RNA sequencing for transcriptomic characterization could be applied for the characterization of several infection states, including staphylococcal bacteremia, Lyme disease, candidiasis, tuberculosis (differentiation between latent and active infection states), and influenza [85]. In addition, another promising application of RNA-seq is in discriminating infectious versus non-infectious causes of acute disease [86].

Although it has been shown to have a promising potential to uncover the “unknown” origin of a disease, it has several obstacles and potential pitfalls when it is applied in the clinical routine, including availability, cost, human resource, and current validity (Figure 1). Indeed, the NGS machine is commonly found in the research laboratory. However, the machine commonly found in the clinical microbiology laboratory might not be an NGS machine. Hence, antibiotic resistance determination relies on the conventional methodology. The workflow of the examination, from sample collection to obtaining the sequencing result, needs to be carefully executed and requires highly trained molecular biology staff. After obtaining the sequence, user-friendly bioinformatics software for the analysis of NGS sequence data is not currently available. Thus, customized bioinformatic pipelines for analyzing clinical NGS data, such as Kraken [76] or SURPI [77], still require highly trained programming staff to develop, validate, and maintain the pipeline for clinical use. In addition, the generated data require a server-class level of computer for computational analysis. Altogether, these will massively increase the cost of examination. In the pathogen detection scenario, the cost for covering the examination is approximately USD 2000–3000 per sample; with this cost, it is easily marked up to >USD 10,000 per charge, which is ridiculously expensive compared to the offered advantages [78]. These challenges might be reduced in the future as the availability of the machine, together with its reagents and highly trained staff, rapidly increases.

Resistome analysis for detecting antibiotic resistance in a clinical sample is still rarely reported. One study showed a promising result for the implementation of resistome mapping using a nanopore sequencing platform. The development of a new pipeline denoted poreFUME showed a promising result of >97% accuracy for annotating ARGs and is considered to be a quick pipeline in the clinical scenario [79]. However, there is still no study comparing the new molecular approach to conventional techniques and describing its reliability to complement or even substitute the current culture-based methodology. Such a study that describes the reliability compared to the current approach is necessary.

## 7. Summary and Outlook

Molecular-based AMR detection shows several advantages compared to current conventional methods. It can predict not only the pathogen’s ARG but also the environment that is related to AMR. The human gut can act as an ARG reservoir; thus, it contains as many ARGs as terrestrial and water environments. Interestingly, antibiotic usage in humans in clinical practice and in cattle for the food industry significantly increased the number of available ARGs and the accumulation of ARGs in the human gut. Therefore, understanding the existence of ARGs in the environment and their potential could lead to better management of infected individuals who need antibiotic treatment. Since the resistance determinant might reside in the environment, the current trend of fecal microbial transplant (FMT) as a new therapeutic approach for several chronic diseases needs appropriate attention for the possibility of transferring ARGs from donors to recipients. Therefore, more detailed screening eligibility for donors is necessary. Currently, the main eligibility criterion for FMT donors is the absence of antibiotic consumption for any reason in the past 3 months. However, there is no criterion for ARG existence in the donor gut [83]. Given the potential transfer of AMR genetic material via the FMT process, ARG screening via metagenomic survey as one of the eligibility criteria is a wise consideration.

The translation of a newly developed methodology from the research laboratory to the clinical laboratory needs to consider analytical validity, clinical validity, and clinical utility [84]. The current molecular-based AMR determination is still in between analytical and clinical validity, and it needs more data and studies to increase its validity. Hence, it is still not recommended to use this particular approach in the routine clinical setting. However, it might be very useful for identifying a particular outbreak of antimicrobial resistance. In addition, determination using metagenomic and metatranscriptomic allows for identifying genera and at least species according to the sequencing method; however, antimicrobial resistance can be detected only at the genomic level. Given this particular difficulty, a good preparation process for the enrichment of DNA available in the specimen is very critical, and currently available sequencers and bioinformatic tools also require highly skilled staff to operate and generate results. Despite these challenges, molecular-based AMR detection is expected to be an interesting subject for research and clinical practice in the coming years.

## Figures and Tables

**Figure 1 antibiotics-11-00654-f001:**
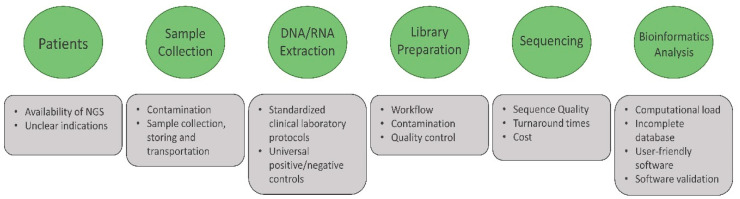
Potential challenges and pitfalls during the implementation of metagenomic and metatranscriptomic profiling of antimicrobial resistance in clinical routines. The potential challenges will be faced by not only the healthcare provider staff but also the patients who need the examination since it is still rarely used. Moreover, the high level of difficulties in executing and maintaining the good quality of the service is still a challenge for implementing this approach as a clinical routine examination.

**Table 1 antibiotics-11-00654-t001:** Microbial characterization technique comparison.

Technique	Explanation	Advantages	Disadvantages
Culture	Bacterial isolation through selective media	Cheap, semiquantitative by counting colonies	Labor intensive, <30% have been cultured
DGGE/TGGE	Gel separation of 16S rRNA amplicons by denaturation-based examination	Fast, semiquantitative, possible to excise bands for further analysis	No phylogenetic identification, PCR bias
T-RFLP	Amplification of labeled primers and digestion of 16S rRNA amplicon. Digested amplicon is separated by gel electrophoresis	Fast, semiquantitative, cheap	Low resolution, no phylogenetic identification, PCR bias
FISH	Fluorescent-labeled probe hybridizes 16S rRNA target. Hybridization occurs and is then quantified by flow cytometry	Phylogenetic identification, semiquantitative, no PCR bias	Probe dependency, inability to identify unknown species
DNA microarrays	Fluorescent-labeled oligonucleotide probes hybridize the complementary nucleotide sequences; laser detection of labeled sequence	Phylogenetic identification, semiquantitative, fast	Cross-hybridization, PCR bias, less applicable for low abundance species
Cloned 16S rRNA gene sequencing	Cloning of full-length 16S rRNA amplicon, Sanger sequencing, and capillary electrophoresis	Phylogenetic identification, quantitative	PCR bias, labor-intensive, expensive, cloning bias
Direct sequencing of 16S rRNA amplicons	Massive parallel sequencing of partial 16S rRNA amplicons	Phylogenetic identification, quantitative, fast, novel bacterial identification	PCR bias
Microbiome shotgun sequencing	Massive parallel sequencing of the whole genome	Phylogenetic identification, quantitative	Expensive, requires high computational power

Abbreviations: DGGE, denaturing gradient gel electrophoresis; FISH, fluorescence in situ hybridization; TGGE, temperature gradient gel electrophoresis; T-RFLP, terminal restriction fragment length polymorphism.

**Table 3 antibiotics-11-00654-t003:** Summary of several discoveries regarding antibiotic resistance by metagenomic approach.

Subjects	Approach	Results	References
Healthy subjects originated from 11 countries (Austria, France, Germany, Iceland, Sweden, China, Japan, USA, Canada, Peru, and Salvador)	Metagenomic characterization and network analysis to establish a comprehensive antibiotic resistome catalog	Vancomycin, β-lactam, tetracycline, macrolide–lincosamide–streptogramin (MLS), bacitracin, and aminoglycoside resistance genes were the most abundant ARG types. Chinese population had the most abundant ARGs.	[75]
Spontaneously delivered infants in Spain	Specific PCR for AMR genes in fecal specimen	Higher β-lactamase-encoding genes detected among received (IAP infants) mothers	[76]
Latin American communities Low-income Latin American communities	Bacterial community characterization and resistance exchange networks from fecal and environmental specimens	Resistomes were associated with bacterial phylogeny structure, and this association was observed across habitats. Several keys of ARGs are associated with MGEs.	[77]
Hunter–gatherers of Hadza people	Functional metagenomic characterization of human fecal specimens	Detected ARGs for synthetic antibiotics among the population, suggesting that the existence of ARGs in the human gut microbiome was independent of commercial antibiotic consumption.	[78]
Three healthy twin pairs in the USA	Characterization of fecal metagenomic and AMR genes	Different ARG characteristics between the babies and their mothers. The resistomes were shared among family members but slightly different across families, suggesting that family-specific shared environmental factors also shape resistome development.	[79]
Individuals from ten different countries (USA, Denmark, Ireland, Spain, France, Sweden, Italy, Malawi, China, and Japan)	Gut resistome comparison between ten different populations	Antibiotic consumption and exposure were strongly associated with the shape of AGRs in gut microbiota. Other factors, such as age, body mass index, sex, or health status, have little effect on shaping AMR potential in human gut microbes.	[80]
Healthy adults and infants from five countries (USA, Japan, Denmark, Spain, and China)	Metagenomic sequencing characterization of human fecal specimens and correlation of antibiotic consumption in humans and animal	Children’s gut resistome characteristics were different compared to their parents. Several ARGs were present, despite no exposure to antibiotics, unusual eating habits, or GI disorder. There was an association between antibiotic use in animals and the enrichment of ARGs in human gut.	[81]
China, Denmark, and Spain	Homology-based prediction and function-based screening of human gut metagenomic sequencing data from public database	Tetracycline ARGs are the most abundant worldwide. The shape of ARG characteristics was determined by country. This shape is likely due to different antibiotic use between those countries.	[82]
Healthy pediatric patients in USA	Functional metagenomic selections with next-generation sequencing	A diverse fecal resistome among healthy children. The detected ARGs among children are independent of antibiotic use. Some ARGs were mobile and had low identity to any known organism, suggesting that the human gut is an important resistance reservoir.	[83]
Remote communities of the Peruvian Amazonas	Specific PCR of ARGs from isolated fecal *E. coli*	High levels of acquired resistance to the oldest antibiotics, such as trimethoprim/sulfamethoxazole, ampicillin, tetracycline, streptomycin, and chloramphenicol, despite the low exposure to commercial antibiotics.	[84]

## Data Availability

Not applicable.

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
