# Peer review of "Antimicrobial Resistance Profile by Metagenomic and Metatranscriptomic Approach in Clinical Practice: Opportunity and Challenge"

_antibiotics, 2022, doi:10.3390/antibiotics11050654_

Round 1

Reviewer 1 Report

Reviewers Comments:

Antimicrobial Resistance Profile by Metagenomic and Metatranscriptomic Approach in Clinical Practice: Opportunity and Challenge

The study discusses the relevant topic of finding alternatives in assessing antimicrobial resistance profiles. Specifically, using metagenomics and metatranscriptomics as a surrogate for culture-based methodologies. Nevertheless, the study requires a considerable amount of revisions spanning across deleting of run-ons, language improvement, extensive citation, and addition of relevant details to improve the intellectual merits and novelty.

Major comments:

ABSTRACT

  1. Page 1, lines 26-44. The abstract does not adequately give a good “snap shot “of the end point of the whole review.
  2. Line 27-28. The statement on how AMR affects government and economics was not explained adequately to the understanding of readers.
  3. The abstract lacks structure, and the review's hypothesis, objective, and significance appear clumped together. Please see the below suggestion:

"The burden of bacteria resistance to antibiotics affects several vital sectors globally, including healthcare, the government, and the economic sector. Resistant bacterial infection is associated with prolonged hospital stays, increased direct cost, and loss of productivity. Current widely performed procedures for identifying antibiotic-resistant bacteria rely on culture-based methodology. However, some resistance determinants are outside the bacterial genome, which could be potentially transferred under antibiotic exposure. Metagenomic and metatranscriptomic approach profiling antibiotic resistance offers several advantages to overcome the limitations of the culture-based approach. These methodologies enhance the probability of detecting resistance determinant genes inside and outside the bacterial genome, and novel resistance genes yet posit inherent challenges in availability, validity, expert usability, and cost. Despite these challenges, such molecular-based and bioinformatics technologies offer an exquisite advantage in improving clinicians' diagnosis and management of resistant human infectious diseases. Hence, this review provides a comprehensive overview of next-generation sequencing technologies, metagenomics, and metatranscriptomics in assessing antimicrobial resistance profiles."

INTRODUCTION

  1. Line 49: “Resistance to antibiotic…”, the “antibiotic” can be made plural since it is not about a particular antibiotic but a general concern of all antibiotics.
  2. The reference stated on line 57 does not provide enough coverage for categorical statement made from line 50-55, as such adequate references should be cited to the statement made.
  3. Page 2, lines 93-94

Although the authors explicitly stated the focus of their review in the gastrointestinal tract (GI), the introduction could have benefited from creating a sub-section to talk about the broad application of the NGS technology in other fields before narrowing it to the GI. The suggested additions will enable other scientists to appreciate the general applicability of these methodologies in tracking antimicrobial resistance in their area and increase this paper's readability and citation.

  1. Global threat to AMR - Page 3, lines 98-119

The information presented by the authors about the global burden is enormous and overlaps with the introductory sections page 2, lines 49-65. While I highly recommend, authors I suggest they delete this section and change the section header to Global burden of human gut-resistant pathogens (since that's the review's focus, as pointed out on page 2, lines 93-94). On the one hand, they could present the mentioned lines in an abridged form and maintain the subsection header and text in the manuscript.  

  1. Table 1: The table is not well organized. If the columns could be made better, it would help. The words of one column run into the other column. The spacing could be increased to separate the columns or lines could be used.
  2. The use of references was limited especially when defining or making key points. It makes the point or definition appear as the writer’s own especially when there are varied opinions with regards to that definition or point.
  3. Structuring of thought processes is key, Consider distinguishing between subject matter on line 119 from line 120, one might misconstrue line 120 to be part of the headed topic; Global impact of AMR
  4. Methodology is quite innovative and gives a solution to acquired resistance whiles patients are on therapy, the short fall is the ability to get results in real time to influence antimicrobial therapy in a clinical setting.
  5. The use of hyphens in a lot of words should be corrected e.g., “in-creased “in line 412
  6. Difference in font alignment between line 453 and 454
  7. Again, the proposed methodology is very resource-dependent and would not be favored in resourced constrained settings, the proposal to have an IT data base of resistant genomes would also come at a humongous cost.

Overall, this was a carefully made analysis of a new trend in combatting AMR, which is an area requiring more work to showcase its advantages over the traditional culture-based approach.

Minor comments:

Please improve language and the appropriate writing of words.

Cite all relevant information comprehensively.

Author Response

To Reviewer : Please see the attachment.

Author Response

To Reviewer: Please see the attachment

Round 2

Reviewer 1 Report

RESPONSE TO REVIEWER 1

Antimicrobial Resistance Profile by Metagenomic and Metatranscriptomic Approach in Clinical Practice: Opportunity and Challenge

The article provides a comprehensive review on metagenomics and metatranscriptomic as a substitute for culture-based approaches in assessing the antimicrobial resistance profile in gastrointestinal tract infections.

The authors have thoroughly addressed all significant comments.

Just a few winding sentences that need to be edited to make easy read and understandable.

Page 3, lines 119-120.

The sentence is unclear, and CDC appears as an organism in this context.

These horrifying bacteria, including drug-resistant non-typhoidal  Salmonella, extended-spectrum beta-lactamase (ESBL)  Enterobacteriaceae, carbapenem-resistant Enterobacteriaceae, and drug-resistant Shigella, and CDC also listed these as their urgent MDR  pathogens.

Please  change “…and CDC also listed these as their urgent MDR pathogens…” to “are also listed by CDC as urgent MDR pathogens.”

Page 3, lines 120-123

The current sentences are difficult to read and remain unclear. Thus whiles the first part of the sentence indicates the link between the condition, organism, and therapy, the second half remains incomplete. Please see suggestions

“In addition, Campylobacter spp [120] has recently developed resistance to major antibiotics such as ciprofloxacin, ampicillin, erythromycin, tetracycline, and trimethoprim-sulfamethoxazole whiles non-typhoidal Salmonella spp has also grown resistance to ampicillin and fluoroquinolones antibiotics used in treating bacterial diarrhea [17].”

Or split the sentences appropriately to make easy read and understanderable.

Page 3, lines 132-137

Please provide references to give credence to the statements.

Reviewer 2 Report

Authors improve the manuscript, they partially rewrite some part of the text as required, but some necessary explanations, evidenced in the first revision report, are included only in the author response, so some inappropriate terms are maintained also in the second version. 

The use of metagenomic in clinical to identify antimicrobial resistance is still unclear; despite this is the main argument of the manuscript.

Round 3

Reviewer 2 Report

The authors sufficiently revise the manuscript.